# Tropospheric Delay Model Based on VMF and ERA5 Reanalysis Data

**Mengtao Zhang [1], Mengli Wang [1], Hang Guo [1,\*], Junjun Hu [1] and Jian Xiong [2,\*]**

[1] School of Information Engineering, Nanchang University, Nanchang 330031, China; 411016620163@email.ncu.edu.cn (M.Z.); 406100210047@email.ncu.edu.cn (M.W.); 411016620148@email.ncu.edu.cn (J.H.)

[2] School of Advanced Manufacturing, Nanchang University, Nanchang 330031, China

\* Correspondence: hguo@ncu.edu.cn (H.G.); xiongjian@ncu.edu.cn (J.X.)

**Abstract:** The global tropospheric zenith delay grid products of VMF1 and VMF3 (Vienna mapping functions) with different resolutions are used to calculate the tropospheric zenith delay of eight IGS (International GNSS Service) stations in China, and the accuracy of the two products under different interpolation methods is analyzed. As a result, the accuracy of utilizing different interpolation methods shows no obvious differences. The interpolation accuracy of the VMF3 grid model is slightly higher than that of the VMF1, and the interpolation accuracy of tropospheric delay is related to the elevation difference of grid points. In addition, according to ERA5 (the fifth generation of the Global Climate Information Analysis data set), the atmospheric stratification tropospheric delay is obtained, and a ZTD (the zenith tropospheric delay) height change grid model is constructed using the least squares exponential fitting method. The accuracy of the model is verified using the tropospheric delay product provided by the IGS. Finally, the constructed ZTD height change grid model is used as ZTD height reduction to solve the problem of large tropospheric delay errors in the VMF interpolation when the height change is large. The model accuracy of URUM station improve from 96.47 mm.to 32.97 mm (65.82%).

**Keywords:** troposphere delay; grid model; grid interpolation; ERA5; VMF

## 1. Introduction

Ionospheric and tropospheric errors are two major error sources for GNSS (global navigation satellite system) applications. Ionospheric errors can be effectively eliminated by multi-frequency signals as the ionosphere is a dispersive medium for microwaves. Unfortunately, tropospheric errors cannot be eliminated by multi-frequency signals, and therefore have a huge impact on positioning accuracy. The tropospheric delay includes the dry and wet components, the former accounting for about 90% of the total contribution delay, which can be effectively corrected using empirical models such as Saatamoinen, Hopfield and UNB3 [1]. Although the wet delay caused by water vapor is small, its change rate is 3–4 times that of the dry delay. Therefore, the tropospheric delay is a main factor restricting ambiguity fixing and subsequently precise GNSS positioning [2,3]. There are usually two methods to deal with tropospheric delay: parameter estimation and model correction. Generally speaking, the accuracy of tropospheric delays obtained by the parameter estimation method is high, but there are also several problems. First, in relative positioning, in the case of small height differences or a short baseline, the difference of tropospheric delays is of course small, and estimating the delay at both stations causes a datum defect. That is why in these cases just the delay difference is solved for as a parameter which solves the problem. On the other hand, modeling the tropospheric delay overcomes the problem of datum defect, but requires an accurate tropospheric delay model. Commonly used tropospheric delay models include Saatamoinen, Hopfield, GMF (global

mapping function), NMF (Niell mapping function) and VMF (Vienna mapping functions), among others. The most commonly used tropospheric model is VMF [4–6].

In the high-precision regional tropospheric delay modeling, finding how to use the tropospheric delay of the reference station to obtain the tropospheric delay of the interpolated station is a key problem. Conventional interpolation algorithms mainly include bilinear interpolation (BIL) and inverse distance weighted interpolation (IDW). There is no obvious difference in the outcome of these interpolation algorithms [7]. Unfortunately, the adaptability is poor in areas with large terrain undulations, and the tropospheric delay error can reach 9 cm when the height difference reaches about 1 km [8]. In [9], Xiaohong Zhang and Feng Zhu et al. compared and analyzed the interpolation effects of different ZTD spatial regression models, and proposed to use the method of height difference constraint to reduce the loss of interpolation accuracy caused by a large height difference. In [10], the authors propose a removal-recovery method based on an empirical model, which can make up for the shortcomings of traditional interpolation methods to a certain extent, but when the height difference is too large, its effect is still not satisfactory.

## 2. Data Analysis

### 2.1. VMF Tropospheric Zenith Delay Product

VMF is a mapping function model established by Vienna University of Science and Technology, and it has released a series of tropospheric delay products in which VMF1 and VMF3 grid products are the most important. VMF1 (Vienna mapping functions) grid products divide the earth surface into $2.5° \times 2.0°$ cells according to geocentric longitude and latitude, and provide global tropospheric zenith delay grid products every 6 h [11]. The data include mapping function coefficients, zenith dry delay (ZHD) and zenith wet delay (ZWD). In addition to grid products, it also provides zenith delay data of global IGS sites, including mapping function coefficients, ZHD and ZWD, and site meteorological data, with the same time resolution as grid products. Through an interpolation calculation of the grid, the grid tropospheric zenith delay can obtain the zenith tropospheric delay of all stations. Therefore, it is more widely used than the site tropospheric delay products. VMF3 is an upgrade product of VMF1, which further improves the grid accuracy on the basis of VMF1 and divides the earth's surface into grids with a $1.0° \times 1.0°$ spacing [12].

### 2.2. IGS Global Troposphere Products

With the improvement of the earth observation network and the increase of observation data, the tropospheric delay transitions from a simple closed mathematical model to a correction model that relies on a large amount of external data [13]. At present, many organizations have provided global tropospheric zenith delay products. These products use different data sources and projection functions, so their accuracy and application areas are different. IGS (International GNSS Service) publishes global tropospheric zenith delay products based on the results of multiple analysis centers, with an accuracy of 4 mm and a wide range of applications [14]. In the process of solving ZTD, it is considered as an undetermined coefficient and solved using six different types of data processing software [15]. Because of the high accuracy of the ZTD issued by IGS, it is often used as a reference true value.

### 2.3. ERA5 Layered Meteorological Data

Layered data sets of air pressure of ERA-Interim and ERA5 are provided by European Centre for Medium-Range Weather Forecasts (ECMWF) for China in 2018. On the vertical section, each grid point of the two models includes the potential height, temperature, pressure, specific humidity, relative humidity and other meteorological elements of 37 isobaric layers, and the height of the top layer is close to 50 km [16]. The difference is that the maximum plane resolution of the ERA-interim stratified pressure data set is $0.75° \times 0.75°$. The temporal resolution of both models is 6 h. The maximum horizontal resolution of the

ERA5 stratified pressure data set is $0.25° \times 0.25°$, and the maximum time resolution is 1 h [17].

### 3. Basic Principle of Calculating Tropospheric Zenith Delay

*3.1. VMF Calculation of Tropospheric Delay*

3.1.1. Tropospheric Delay Elevation Correction

The following equation is used to reduce the grid ZHD to the height of the station point (site), and the horizontal location of the station point is not changed compared to the grid point. The grid point height is $h_g$ and the site height is $h_s$. The tropospheric delay grid elevation needs to be reduced to the station elevation.

$$D_H(h_s) = D_H(h_g) - 2.277 \times 10^{-3} \times \frac{g \times P(h_g) \times (h_s - h_g)}{R \times T(h_g)} \tag{1}$$

where $D_H(h_s)$ and $D_H(h_g)$ are the dry delay of the site and grid points, respectively; $h_s$ and $h_g$ are station height and grid height (unit:m); $T(h_g)$ is the temperature corresponding to the grid point (unit:K); $p(h_g)$ is the air pressure corresponding to the grid point (unit:hPa); temperature and pressure calculated by means of the GPT (global pressure and temperature) model or standard atmosphere; $R$ is a constant number of gases, $R = 287.054 \mathrm{J}/(\mathrm{kg} \cdot \mathrm{K})$; $g$ is the gravity constant, $g = 9.784 \mathrm{\ m/s^2}$. Since the ZWD caused by water vapor has the characteristics of large rate of change and randomness, the exponential attenuation function is selected to correct ZWD for the elevation [11].

$$D_W(h_s) = D_W(h_g) \times e^{-(h_s - h_g)/2000} \tag{2}$$

where $D_w(h_s)$ is the ZWD at the reference station (unit:mm); $D_w(h_g)$ is the ZWD of the grid point (unit:mm); $h_s$ is the height of the station (unit:m); $h_g$ is the height of the grid point (unit:m); $e$ is the Euler number equal to approximately 2.71828.

3.1.2. Tropospheric Delay Grid Plane Interpolation Scheme

In order to analyze the effect of the horizontal interpolation strategy on the results, BIL and IDW are used respectively. The mathematical model of the BIL method is as follows:

$$p = (\lambda - \lambda_{00})/\Delta\lambda, q = (\varphi - \varphi_{00})/\Delta\varphi \tag{3}$$

where $0 \leqslant p < 1, 0 \leqslant q < 1$, $\lambda$ and $\varphi$ are the longitude and latitude of the interpolation point; $\lambda_{00}$ and $\varphi_{00}$ are the longitude and latitude of the lower left corner of the grid cell, where the interpolation point is located. At first, points A, B or C, D are linearly interpolated. Then, point Z is linearly interpolated, and the $Z(\lambda, \varphi)$ value of point Z is calculated as:

$$Z(\lambda, \varphi) = Q_{00} \times Z_{0,0} + Q_{10} \times Z_{1,0} + Q_{01} \times Z_{0,1} + Q_{11} \times Z_{1,1} \tag{4}$$

$$\begin{cases} Q_{00} = (1 - p) \times (1 - q) \\ Q_{01} = (1 - p) \times q \\ Q_{10} = p \times (1 - q) \\ Q_{11} = p \times q \end{cases} \tag{5}$$

where $Z_{0,0}, Z_{1,0}, Z_{0,1}, Z_{1,1}$ are ZTD values of the four grid points around the point P, respectively shown in Figure 1, which can be obtained jointly by Equations (1) and (2); $Q_{00}, Q_{01}, Q_{10}, Q_{11}$ are the corresponding factors. The mathematical model of inverse distance weighted interpolation (IDW) reads as follows:

$$\lambda_i = \frac{1/d_i^x}{\sum_{i=1}^{n} 1/d_i^x} \tag{6}$$

$$S_{site} = \sum_{i=1}^{n} \lambda_i S_{grid,i} \tag{7}$$

where $n$ is the total number of grid points around the station; $d_i$ is the distance between the $i$ grid point and the station; $\lambda_i$ is the weight of the corresponding mesh point; $x$ is the power value of inverse distance weighting, generally 1.3; $S_{site}$ and $S_{grid,i}$ are tropospheric delay quantities at the station and the ith grid point, respectively.

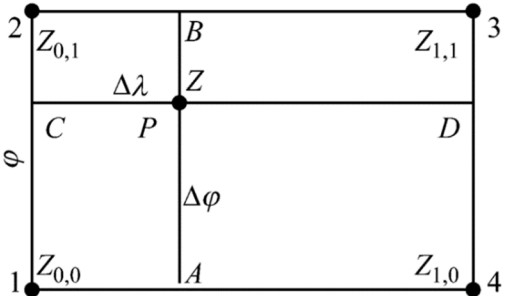

**Figure 1.** Interpolation of 4 points around.

*3.2. The Principle of ERA5 Data Set Calculation to the Path Layer Delay*

The Saatamoinen method and the integration method are commonly used to calculate ZTD from ECMWF reanalysis data. In [11], Kouba points out that the accuracy of the integration method to calculate ZTD is better than the Saatamoinen method. Therefore, this paper uses the integration method to calculate the ZTD of ERA5 data. The applied formula for calculation of the ZTD reads:

$$ZTD_i = 10^{-6} \times \int N ds = 10^{-6} \times \sum_{1}^{i} N_i \times \Delta s_i + ZTD_{top} \tag{8}$$

where $ZTD_i$ is the zenith tropospheric delay on the isobaric surface of the $i$th layer calculated from the top layer; $ZTD_{top}$ is the ZTD above the top layer; $N$ is the total atmospheric refractive index; $N_i$ is the atmospheric refractive index of the ith layer; $\Delta s_i$ is the height difference between level $i$ and level $i$-1, with the topmost level $\Delta s = 0$. Since there is almost no wet delay above the top level, the $ZTD_{top}$ can be replaced by the tropospheric zenith dry delay (ZHD) above the top layer, calculated by the Saatamoinen model. The calculation method of atmospheric refractive index is as follows:

$$N = \frac{k_1(P - e)}{T} + \frac{k_2 e}{T} + \frac{k_3 e}{T^2} \tag{9}$$

$$e = h \times P / 0.622 \tag{10}$$

Among them:

$$k_1 = 77.604 \ \text{K/P}_a$$

$$k_2 = 64.79 \ \text{K/P}_a$$

$$k_3 = 337600 \ \text{K}^2/\text{P}_a$$

where $P$ is atmospheric pressure (unit:hPa); $e$ is water vapor pressure (unit:hPa); $T$ is temperature, and $h$ is specific humidity. Because the height system of ECMWF is geopotential height, while the height system of GPS station is geodetic height, in order to facilitate comparative analysis, it is necessary to convert the geopotential height into geodetic height. In [18], Chen M and Chen J.P. et al. introduced specific elevation system conversion meth-

ods. After unifying the elevation system, the atmospheric refractive index of each layer can be calculated by using the meteorological parameters of each layer. In order to improve the accuracy of calculating ZTD by the integration method, the meteorological parameters of each layer should take the mean value of the meteorological parameters of the two adjacent isobaric surfaces [19]. Only ZTD on the ECMWF barometric stratified data grid can be obtained by the integration method. To obtain ZTD of the GNSS station location, ZTD of the four nearest grid points around the station should be obtained by bilinear interpolation. Because the height of the GNSS station and its surrounding four grid points from ECMWF reanalysis data are different, it is necessary to reduce ZTD calculated from ECMWF reanalysis data to the same height as the GNSS station. Next, bilinear interpolation is used to perform horizontal interpolation. In [20], when ZTD is scaled in the vertical section, the second order polynomial is used to fit the change of ZTD with height. This method will cause the precision loss of ZTD in the process of fitting and interpolation.

In this paper, ZTD of the grid point height at the station is directly calculated by the integration method, and then ZTD of the GNSS station height is interpolated by the bilinear interpolation level [21]. When calculating ZTD of the station height by integration, this method needs to obtain the meteorological elements of the grid point at the station height. When the height of the station is higher than the lowest level of the meteorological reanalysis data, the meteorological elements of the station height are obtained by linear interpolation of the meteorological elements of the two adjacent layers of the station height. When the height of the station is lower than the lowest level of the meteorological reanalysis data, the meteorological elements of the lowest level need to be extrapolated [22]. In [23], specific methods for the derivation of meteorological elements are provided.

### 3.3. Model Accuracy Test

In this paper, the average deviation (Bias) and root mean square error (RMSE) are used as the model accuracy evaluation criteria [24], and the calculation formula is as follows:

$$Bias = \frac{1}{D} \sum_{d=1}^{D} \left( ZTD_d^{Model} - ZTD_d^{IGS} \right) \tag{11}$$

$$RMSE = \sqrt{\frac{1}{D} \sum_{d=1}^{D} \left( ZTD_d^{Model} - ZTD_d^{IGS} \right)^2} \tag{12}$$

where $ZTD_d^{Model}$ is the value calculated by VMF model interpolation in this paper; $ZTD_d^{IGS}$ is the true value provided by the tropospheric product released by IGS; $D$ represents the total number of observed values; Bias measures the average deviation between the calculated value of the model and the true value; RMSE measures the reliability of the model.

## 4. Acquisition and Analysis of VMF Experimental Data

### 4.1. Source of Experimental Data

It primarily uses VMF's grid data and grid point elevation to calculate tropospheric delay. VMF can be downloaded from https://vmf.geo.tuwien.ac.at/index.html (accessed on 15 April 2022). We accessed the data on 15 April 2022. This experiment uses VMF1 and VMF3 grid data in 2019 and 2020 (from 1 January 2019 to 31 December 2020), and the grid point elevation data uses $2.5° \times 2.0°$ grid elevation and $1° \times 1°$ grid elevation data. In this experiment, the coordinates of eight IGS stations in China are selected as the solution coordinates of the VMF model to solve the troposphere. The distribution of the stations is shown in Figure 2.

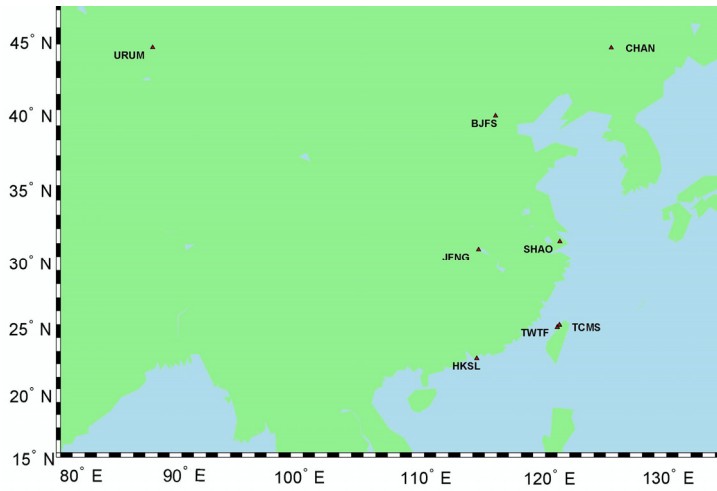

**Figure 2.** Station distribution map.

### 4.2. Experimental Scheme

Combining two VMF datasets and different interpolation methods, the calculation schemes 1–4 are composed as follows: the first scheme uses VMF1 data and the bilinear interpolation method (VMF1 + BIL); the second scheme uses VMF1 data and inverse distance weighted interpolation (VMF1 + IDW); the third scheme uses VMF3 data and bilinear interpolation (VMF3 + BIL); and the fourth scheme uses VMF3 data and inverse distance weighted interpolation (VMF3 + IDW). The experimental results are shown in Figures 3 and 4.

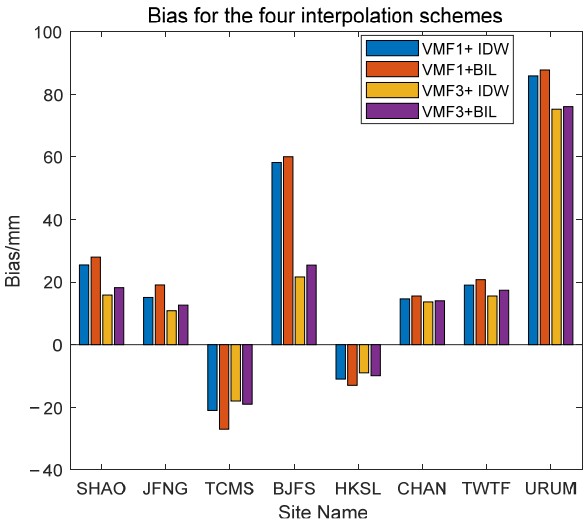

**Figure 3.** ZTD average deviation under the four solution schemes.

As shown in Figure 3, the ZTD at the stations obtained using four experimental schemes is greater than the true value, except for the TCMS and HKSL stations. It can be seen from Figures 3 and 4 that except for URUM and BJFS, the interpolation accuracy obtained by the four schemes at each station is less than 40 mm, BIL is slightly better than IDW, and ZTD obtained by VMF3 grid is slightly higher than VMF1. The interpolation accuracy is lower when using VMF1 to interpolate the ZTD at BJFS, and its average root mean square error is 81.785 mm. Similarly, the interpolation accuracy is lower when using VMF3 and VMF1 to interpolate the ZTD at URUM, and its average root mean square error is 96.47 mm.

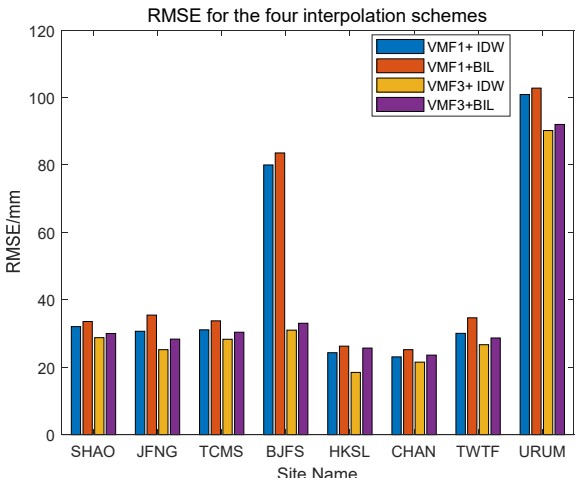

**Figure 4.** ZTD root mean square error under four solution schemes.

In order to verify the relationship between the tropospheric solution accuracy and the elevation of the four grid points around the station, the elevation difference of the four points around each station is calculated. Information on the stations is shown in Table 1. The calculation formula for elevation difference is as follows [25]:

$$\Delta h = \sqrt{\sum_{i=1}^{4}(h_i - h)^2/4} \tag{13}$$

where $h$ is the elevation of the site and $h_i$ is the elevation of the four grid points (unit: m). The height difference between the IGS reference station and the surrounding four grid points can be calculated using Equation (13), as shown in Figure 5.

**Table 1.** Station information.

| Station Name | Longitude/(°) | Latitude/(°) | Height/m |
|:---:|:---:|:---:|:---:|
| SHAO | 121.20 | 31.10 | 22.11 |
| JFNG | 114.49 | 30.51 | 71.30 |
| TCMS | 120.99 | 24.80 | 77.30 |
| BJFS | 115.89 | 39.61 | 87.40 |
| HKSL | 113.93 | 22.37 | 95.26 |
| CHAN | 125.44 | 43.79 | 268.30 |
| TWTF | 121.16 | 24.95 | 203.10 |
| URUM | 87.6 | 43.81 | 749.54 |

*4.3. Conclusion Analysis*

From Table 1, it can be shown that the elevations of the CHAN, TWTF and URM stations are larger, and the interpolation accuracy of the CHAN and TWTF stations is higher, with the root mean square error less than 40 mm. Therefore, there is no direct relationship between station elevation and interpolation accuracy.

From the elevation difference in Figure 5, using the VMF1 grid, the elevation differences of URUM and BJFS are 1069.34 m and 691.35 m, respectively, and using the VMF3 grid, the elevation difference of URUM station is 901.52 m, which is relatively large. According to the data, Beijing is located in the North China Plain, adjacent to the Yanshan Mountains in the northwest, with an elevation difference of 1200 m; Urumqi is at the north foot of the Tianshan Mountains. To the south is the towering Tianshan Mountains, and to the north is the vast Junggar Basin, with an elevation difference of 1000 m. In the above terrain, the elevation difference between the four grid points around the selected survey

station and the interpolation station is large, so there will be large errors in the tropospheric delay elevation correction, resulting in the low accuracy of the interpolation results.

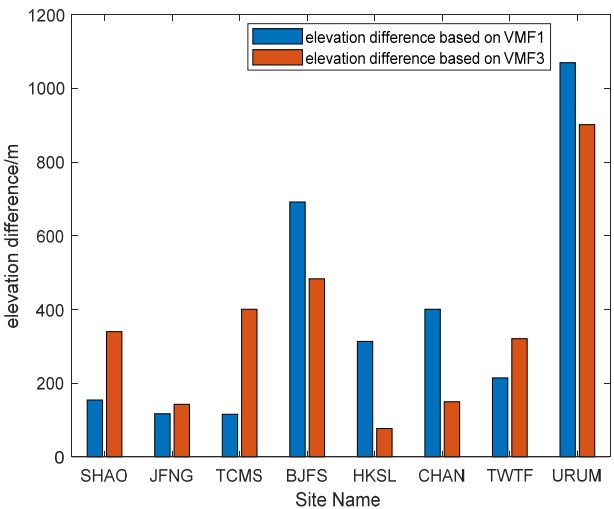

**Figure 5.** Elevation difference.

## 5. Building ZTD Height Change Grid Model Based on ERA5

There are many global tropospheric delay models, but no high-precision model for ZTD with altitude correction [26]. In this paper, the monthly products of ERA5 barometric layer in 2018–2020, including temperature, pressure, humidity and height, are used to establish the model of ZTD varying with height in some regions of China. The accuracy of the model of ZTD varying with height is verified using ZTD provided by IGS in 2021 and ZTD calculated by ERA5.

### 5.1. Selection of Fitting Model for ZTD Variation with Height

Since the ZTD in the region above 10 km from the ground is very small, the ZTD below 10 km is selected for discussion and analysis of the ZTD variation with height. The relationship between ZTD and height calculated by ERA5 below 10 km in 2018–2020 is shown in Figure 6.

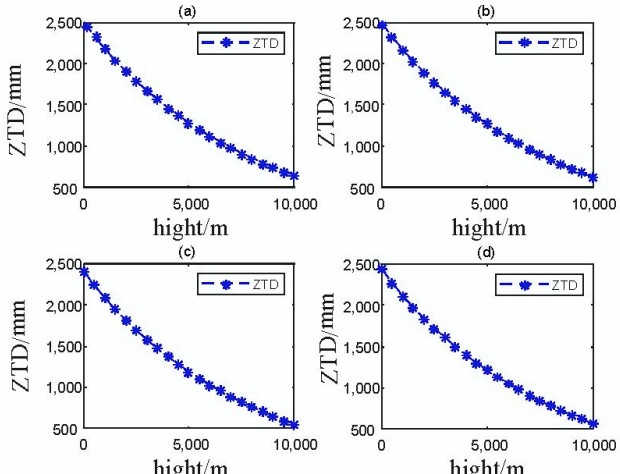

**Figure 6.** Scatter diagram of ZTD changing with height (**a**) (87°, 22° N) (**b**) (87°, 43° N) (**c**) (126°, 22° N) (**d**) (126°, 43° N).

As can be seen from Figure 6, ZTD below 10 km clearly shows an exponentially decreasing form with height change, which can be fitted using an exponential model, and

an exponential model of ZTD with height change for some regions in China (70°~140° W, 15°~55° N) is constructed using the inverse ZTD from the 2018–2020 ERA5 reanalysis data.

### 5.2. Least Square Exponential Fitting

The least squares fitting principle is to determine the coefficient of the exponential function by taking the least squares sum of the residuals between the dependent variable of the fitting exponential curve and the observed value as the optimal objective [27–29]. According to the given data, combined with the scatter diagram, it is possible to determine whether there is an exponential relationship and to solve the exponential fitting function coefficient. The exponential equation is as follows:

$$y = \alpha e^{\beta x} + \varepsilon \tag{14}$$

The sum of the squares of residuals for exponential fitting is as follows:

$$min f(\alpha, \beta) = \sum_{i=1}^{n} \left[ y_i - \left( \alpha e^{\beta x_i} + \varepsilon \right) \right]^2 \tag{15}$$

where $\alpha$ and $\beta$ are the least squares fitting coefficients.

### 5.3. Model Accuracy Verification

To verify the experimental results, the fitted ZTD deviation and root mean square error are calculated. In this case, the ZTD is obtained using the integration of the 2021 ERA5 barometric stratified data set as the reference true value, and the least-squares exponent is used to fit the ZTD with an accuracy of mm level. The average deviation on the 37-layer isobaric surface for all grid points is 6 mm, and the average root mean square error is 8.5 mm. The root mean square error is shown in Figure 7.

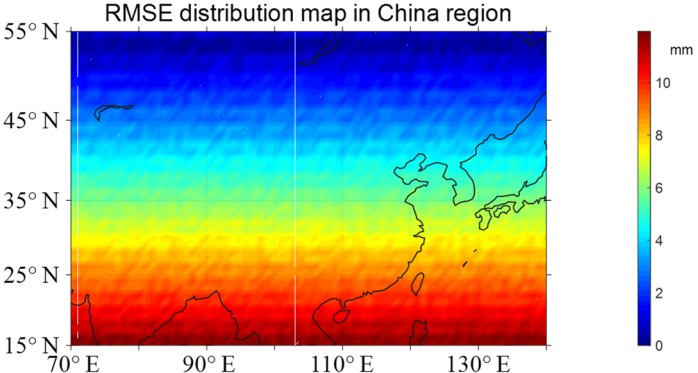

**Figure 7.** Distribution of root mean square error in China.

It can be seen from Figure 7 that the least squares index fitting ZTD has high accuracy in China, with the root mean square error within 12 mm. The fitting accuracy is related to the latitude. The higher the latitude, the higher the fitting accuracy. According to the data, the air in the low latitude area contains more water vapor, and the climate is relatively humid, so it is difficult to accurately estimate ZWD.

## 6. Correct VMF Interpolation Model Based on Height Change Grid Model

### 6.1. Elevation Reduction of Grid Model Based on ERA5 Height Change

By VMF analysis in Section 4, the elevation difference between the selected station and the surrounding four grid points is large, so there will be large errors in the tropospheric delay elevation correction, which will lead to the low accuracy of the interpolation results. Therefore, the elevation change grid model is used for tropospheric delay elevation

correction. First, the tropospheric delay grid elevation needs to be reduced to the station elevation.

$$D(h_s) = D(h_g) + \alpha_s e^{\beta_s h_s} - \alpha_g e^{\beta_g h_g} + \varepsilon_s - \varepsilon_g \qquad (16)$$

where $D(h_s)$ and $D(h_g)$ are ZTD of the site and grid point respectively; $h_s$ and $h_g$ are station height and grid height (unit:m); $\alpha_s, \beta_s, \varepsilon_s$ are the fitting coefficients of the elevation change grid model at the interpolation points; $\alpha_g, \beta_g, \varepsilon_g$ are the fitting coefficients of the elevation change grid model at the VMF grid points.

### 6.2. Design and Analysis of Experimental Scheme

This experiment is based on the experiment in Section 4. Equation (16) is used when the ZTD at the grid point is normalized to the same height as the station. The experimental results are shown in Figures 8 and 9.

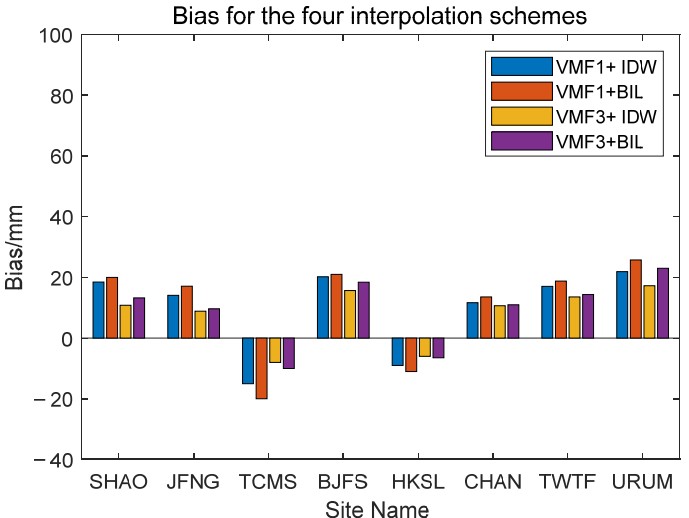

**Figure 8.** ZTD average deviation after correction of elevation mode.

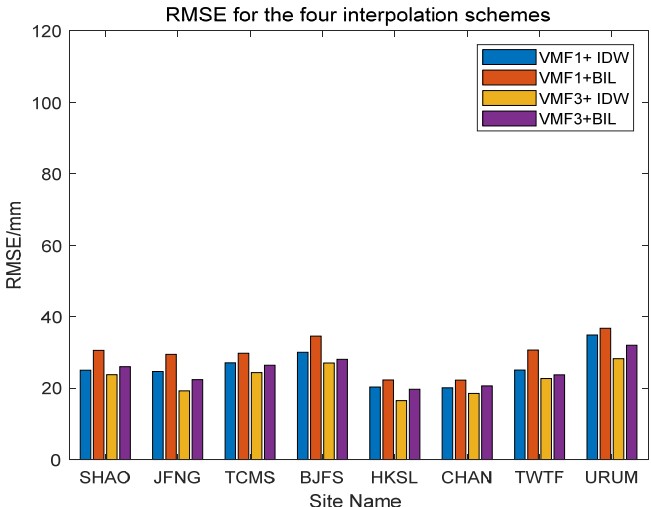

**Figure 9.** ZTD root mean square error after correction of elevation mode.

From Figures 8 and 9, it can be seen that VMF interpolation model is corrected by using the height-varying grid model to achieve better results, and the accuracy of the internal and external compliance of all four models is improved. Among all the stations, the URAM and BJFS stations showed the largest improvement in interpolation accuracy. From Figures 3 and 8, it can be seen that using VMF1 to interpolate the ZTD at BJFS, its

average deviation is reduced from 64.12 mm to 25.62 mm; the average deviation of ZTD at URUM is reduced from 86.22 mm to 26.98 mm. From Figures 4 and 9, it can be seen that using VMF1 to interpolate the ZTD at BJFS, its average root mean square error of ZTD at BJFS is reduced from 81.785 mm to 32,285 mm; the average root mean square error of ZTD at URUM is reduced from 96.47 mm to 32.97 mm. The interpolation accuracy improvement of other stations is not too obvious, and the average improvement is about 5 mm.

## 7. Conclusions

In this paper, the accuracy of VMF model interpolation into the zenith troposphere delay is experimentally analyzed, and a ZTD height change grid model is constructed based on ERA5 reanalysis data. A ZTD height change grid model is proposed to correct VMF interpolation model, and the accuracy of the model is verified.

(1) Using the VMF grid model to interpolate the tropospheric delay, and using the function of the troposphere delay changing with the elevation to correct the tropospheric delay will produce a large error. The greater the elevation difference, the greater the error. The elevation difference between the URUM station and its surrounding grid points is the largest (about 1000 m), and the resulting tropospheric delay error is also the largest (about 100 mm).

(2) The tropospheric delay is obtained using ERA5 reanalysis data integration. According to the analysis, the ZTD below 10 km in China clearly shows an exponentially decreasing form with the altitude change.

(3) The least squares exponential fitting method is used to establish a ZTD height change grid model in China, and the ERA5 integration is used to obtain ZTD delay to verify the accuracy of the model. The root mean square error is less than 12 mm. The fitting accuracy is related to the latitude. The higher the latitude, the higher the fitting accuracy. The reason for the lower fitting accuracy at low latitudes is that the air at low latitudes contains more water vapor and the climate is relatively wetter, making it difficult to estimate ZWD accurately.

(4) Through experimental analysis, using the ZTD height change grid model as the ZTD height reduction method can effectively reduce the error caused by the tropospheric delay of VMF interpolation when the height difference is large. The VMF interpolation accuracy improvement at URAM is the largest, with an improvement of 63.5 mm in root mean square error.

**Author Contributions:** Conceptualization, M.Z. and H.G.; methodology, M.Z.; software, M.W.; validation, H.G., J.X. and J.H.; formal analysis, M.Z.; investigation, M.Z.; resources, M.W.; data curation, H.G.; writing—original draft preparation, M.Z.; writing—review and editing, H.G.; visualization, M.Z and M.W.; supervision, H.G.; project administration, J.X.; funding acquisition, H.G. and J.X. All authors have read and agreed to the published version of the manuscript.

**Funding:** This research was funded by the National Natural Science Foundation of China (Grant Nos. 41764002; 62263023; 62161022).

**Institutional Review Board Statement:** Not applicable.

**Informed Consent Statement:** Not applicable.

**Data Availability Statement:** Not applicable.

**Acknowledgments:** We thank International GNSS Service for the reference station data sets, Vienna University of Science and Technology for Vienna Mapping Functions data (VMF), and the European Centre for Medium-Range Weather Forecasts (ECMWF) for the integration of the fifth generation of global climate in the analysis data set (ERA5).

**Conflicts of Interest:** The authors declare no conflict of interest.

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
