# Peer review of "Tropospheric Delay Model Based on VMF and ERA5 Reanalysis Data"

_applsci, doi:10.3390/app13095789_

Round 1

Reviewer 1 Report

The paper deals with the use and evaluation, under various interpolation algorithms, of the zenith delay grid products VMF1 and VMF3 to estimate the tropospheric zenith delay for a number of IGS stations in China. Moreover, the ERA5 model is used to estimate a height change grid model. In general the paper is interesting and scientifically sound, but it
needs attention to the use of English. In quite a few places, the writing is not the one expected for a scientific journal, while the meaning is difficult to follow.

As an example (lines 19-20, quote): "Ionospheric errors can be effectively eliminated by multi-frequency signals, but tropospheric errors cannot be eliminated
by multi-frequency signals, which have a great impact on positioning accuracy". This should be rephrased as: "Ionospheric errors can be effectively eliminated by multi-frequency
signals, but tropospheric errors cannot, hence the latter have a great impact on positioning accuracy."

Also, (lines 21-22, quote): "The tropospheric delay includes dry delay and wet delay, and dry delay accounts for about 90 % of the total delay, which can be accurately
corrected by using...." should be rephrased to "The tropospheric delay includes the dry and wet components, the former accounting to about 90 % of the total contribution delay, which can
be effectively corrected using ....".

Moreover, the use of the definite and indefinite article is not proper or not present as all, which makes the natural flow of reading difficult. Finally, several sentences are large
so that their meaning is lost.

Therefore, the text needs a general re-writing with attention to the use of English, so as to improve its quality and readability.

In terms of more specific comments, a comprehensive list is provided below:

1) Line 44 and LIne 47. The authors mention "The literature". This is a very unconventional way of citing related work and doing a literature review. It should be cited as e.g. " In [6]...."

2) The list of references is quite limited to work performed by Chinese researchers and to journals that mostly reside and are published in China. This is a significant limiting factor for the paper, as well-established geodetic and GNSS journals are not reviewed and the work performed by researchers during the last twenty years, which appear on these flagship
journals are omitted. To the reviewers point view, the list of references needs a complete revision, as it missed quite a few of papers of the pioneering work on the ZTD modeling and
estimation. It is understandable that the authors, coming from Chinese universities find more accessible papers from Chinese colleagues, but limiting the literature to that
is not proper from a scientific point of view. This is even more evident from the fact that even the models evaluated (VMF1 and VMF3) are not given their citation. This should be
corrected and addressed by the authors in the revised version of the paper.

3) Line 79. The same, as in comment 2 above, holds in this paragraph for the ERA5 model, which is not cited at all.

4) Lines 98-101. This sentence is very complicated and the reader is lost. What the authors intend to say is that they used an exponential decay model for the wet tropospheric correction. Instead of that they complicate things and make the sentence difficult to follow. Please re-write.

5) What other options does one have available to estimate both the wet and dry components? A critical review of the various analytical models and a justifiaction of the selections
should be given.

6) Line 121: ....from ECMWF

7) Line 136: The authors use again an unconventional notation as (quote) " is shown in document[14]". Please use a better phrasing.

8) Line 141. What do you mean that the GNSS station height is inconsistent with the ECMWF reanalysis data?

9) Line 147. Why is a second order polynomial appropriate to approximate the downward continuation of ZTD with height? This presupposes that the ZTD varies quadratically
with height. Is that so?

10) Section 3.3. The authors use a bias and rms statistic measure to conclude on the agreement between the predicted and provided by IGS ZTD values. Of course these measures
are pretty standard and they can provide a first outlook on the results achieved, but I would suggest to also derive some corellation measure, the mean mean squared deviation, the
concordance correlation coefficient, the total deviation index, etc., among others, as they can provide some inside as well.

11) Sections 4.1 and 4.2. What is the number of samples (parameter D) for the stations, based on which the authors calculate the statistics?

12) In section 4.2 the analysis of the results is pretty basic and the authors do not try to interpret why they get the results they get. For instance, stations TCMS and HKSL
have a mean that has a reverse sign compared to the other stations. Why is that? How is that quantified? Does the station height has to do something with that? The station
information is not provided by the authors. Also, they estimate a mean and and rmse. In this context, the std would be better as the rmse is biased and reflects, as seen
from the values in Figures 2 and 3 from the mean. The quantified and critical analysis of the results is also important in view also of comment 10) above.

13) Section 6.2. The results achieved with the proposed ZTD estimation based on the fit to the ERA5 height change model, in order to avoid interpolation from grid points
with essentially several hundred meters height difference, are worthwhile. Again, the authors do not make a comparative assessment with the results achieved in Section 4.2
with the more classical approach. Also, why this exponential model and not another one? What is the goodness of fit? Did you estimate any coefficient of determination
and/or condition number (when you solve this parametric model fit)?

14) In the conclusion section, the authors provide a numbered list of their main findings. I would suggest to provide a more comprehensive text, where there will be a
discussion of the results and an interpretation of why you reach these results.

Reviewer 2 Report

Dear Authors

in your manuscript you describe a method to interpolate ZTD from VMF1 and VMF3 grids and you propose a kind of correction technique for dealing with significant height differences between your observ ation stations and the grid points. In general this topic is quite interesting but I have identified a lot of technical issues with your text. Moreover the English grammar is far from acceptable and I give you the major hint to forward the text to an English native speaker for corrections. However fins below A) a list of the technical issues/questions and also B) a list of grammar hints. Please note that the grammar hints are by far not conclusive and the list can be extended by far. My points refer to the line numbering scheme and the equation numbers of your paper

A) technical hints/questions

0) The titel of your manuscript does not really reflect the major topic of the paper; you focus on a correct interpolation of the VMF grids to observation sites

1) Equation (1); I assume this equation is used to correct the ZTD at the grid point heights to the ZTD at the grid point (but for the stations height); If yes so please note this clearly in the text - or provide another reasonable explanation; explain further if you use exact these corrected values as input for equation (4)

2) equation (2); this equation is not reasonable using the given explanations;

 explain the quantity 'e' : I assume its the water vapor pressure

 if you enter the height difference as the power of e as noted in meters, the equation gives no reasonable results; shall hs and hg in units [km] ??

 give a correct explanation of the quantities

3) sometimes  you use in your manuscript the acronym 'BI' and sometimes 'BIL' for bilinear interpolation

4) Following Figure 5 it looks like that the elevation difference between mean of grid points and the observing station is larger for VMF3 than for VMF1; Thats slightly astonishing as the grid size for VMF3 is smaller and the grid heights should be closer to the station height. Is figure 5 really essential as you corect anyway the first of all the grid ZTDs for height differences to the station height ?

5) equation (16) and equation (1): in both cases you note that 'the tropospheric delay grid elevation needs to be reduced to the stations height'

but in the formulas you reduce the ZTD (and not the elevation) for height differences between grid points and observation site; explain clearly

Thats essential for understanding equation (16)- do you apply the fitted corrections after you have already corrected the grid ZTDs to the station height or do apply you exponential corrections on the grids ZTDs already in the height of the observation site?

6) a general comment to the References section

There is mostly Chinese literature cited; you have also to add international papers to this list . The most striking ommision is that you have not even cited papers of the providers of the VMF grids -> see VMF webpages

 References : Boehm et al.

There are also some typos in your References list; There is also a bulk of papers dealing with the VMF grids issued by the IGS community

Nr4 : 'Posirtioning'

Nr10: Which Journal is called just 'Global Positioning System'?

Nr 13: 'Lea-st'

B) grammar hints

1) line 4

'.. and the accuracy of the two products .. methods is analyzed.'

2) next sentence

'As a result the accuracy utilizing different interpolation methods shows no obvious differences.'

3) l5

'.. of the VMF3 grid...'

4) l6 its unclear what you understand under

'.. the elevation reduced accuracy..'??

5) l9

'.. and a ZTD (...'

6) l11

'.. provided by the IGS. Finally...'

7) l12

'.. delay errors in ...'

8) keywords; check your keywords

I assume 'Least squares' and 'elevation difference' are no meaningful keywords; Instead of 'Troposphere' introduce 'Tropospheric Delay'

9) l20,21

'.. multi-frequency signals as the Ionosphere is a dispersive medium for microwaves. Unfortunately, tropospheric errors cannot... signals and have therefore a huge impact on...'

10) l25

what do you mean with 'the fixed' ? an ambiguity fixed solution?

11) l28

'.. of tropospheric delays obtained...'

12) l29-33

a more technical than grammar hint

in relative positioning in case of small height differences or a short baseline the difference of tropospheric delays is of course small and estimating the delay at both stations causes a datum defect. That's why in theses cases just the delay difference is solved for as a parameter which solves the problem.

13) l33

re-phrase : 'The correction of tropospheric delay model does not have the above defects..'

14) l36

'The most commonly used tropospheric model is VMF.

15) l38-40

 re-phrase the sentence 'In the high-precision..'

16) l41

'There is no obvious difference in the outcome of these interpolation algorithms[4]. Unfortunately, the adaptability...'

17) l47

'.. caused by a large height..'

18) l48

'.. based on an empirical model...'

19) l48

'.. difference is too large, its...'

20) l56

'.. divide the earth surface in 2.5x2.0 cells according..'
21) l61

'.. Through interpolation the grid ...'

22) l62

'.. delay of all stations. Therefore it is more...'

23) l64

'.. and divides the earth's surface in grids with a 1.0x1.0 spacing.'

24) l74

The sentence 'ZTD product.. ' has to be re-phrased

25) l79,80

'.. and ERA5 are provided by the European ... (ECMWF) for China in 2018.'

26) l84-86

'... 0.75x0.75. The temporal resolution of both models is 6h. The maximum horizontal resolution of the ERA5...'

27) l90 , header of chapter 3.1.1

'Tropospheric...'

28) in line 93 you mixed the sequency of grid height and station height

29) l96

'.. temperature and pressure calculated by means of GPT (...'

30) l98-100

 I assume from the sentence what you want me to tell but please re-phrase

this sentence to provide a clear expression

31) l101

'.. to correct ZTD for the elevation...'

32) l106

'.. the effect of the horizontal interpolation..'

33) l107

'.. model of the BI method...'

34) l109,110

'.. left corner of the grid cell, where... is located. At first, points A,B or C,D are linearly interpolated. Then,...'

35) l112

here you should be consistent with the graph -> use point Z instead of 'P'

36) l113

..Q11 are the corresponding..'

37) l114

'.. (IDW) reads as follows:'

38) l117

'.. are tropospheric delay quantities at ...'

39) l121

'.. calculate ZTD from ECMWF ...'

40) l122

'.. Saastamoinen method. So this paper...'

41) l123

'.. ERA5 data. The applied formula for calculation of the ZTD reads:'

42) l128

'.. above the top level. The ZTD...'

So I stop here (due to lack of time), although also the upcoming chapters have to be checked for grammar and typos

best regards

Reviewer 3 Report

In this paper, the authors constructed a ZTD (Zenith Tropospheric Delay) height change grid model based on ERA5 reanalysis data, and the accuracy of VMF model interpolation is improved. There are some suggestions listed as follows:

1. The introduction section can be improved, more related research results should be reviewed.

2. In the text below eq. (1), "hs and hg are grid height and station height" --> "hs and hg are station height and grid height". A similar mistake appears in the text below eq. (16).

3. In the text above eq. (13), "the elevation difference of the four points around each station and the station is calculated" --> "the elevation difference of the four points around each station is calculated".

Round 2

Reviewer 1 Report

The authors have provided some arguments on the first review and corrected the paper accordingly. Nevertheless, they have not updated the text with respect to the needed discussion on the achieved results. These are points #14 and #15  on their response to reviewer comments. In Sections 4.2/4.3 and 6.2 a more critical analysis of the achieved results is needed. Why are these results achieved? Why do they differ by station? Does the height of the stations have to do something with that? These issues, pointed out in the original review, have not been addressed.

Reviewer 2 Report

Dear Authors

your manuscript has been definitely improved during the past revision cycle;

most of my grammar hints were accepted, so I restrict my comments below to very few remaining cases. Nevertheless, there is still some room for an additional grammar check;

1) page 1 Introduction, line9 ; unfortunately the sentence is not so much better than before

I propose 'Therefore, the tropospheric delay is a main factor restricting ambiguity fixing and subsequently precise GNSS positioning [2,3].'

2) page 1 , 3 lines from bottom

substitute the sentence 'Using model...' by

'On the other hand, modeling the tropospheric delay overcomes the problem of datum defect but requires an accurate tropospheric delay model.'

3) page 3 chapter 3

  Concerning my point 1 in the last revison ou noted in your reply letter that

  its a good suggestion to explain at the start of chapter 3 that formula (1) is just to reduce the grid ZHD to the height of the station point (but still located in the horizontal grid. Unfortunately I still miss such an introductionary sentence to chapter 3 in your manuscript

4) equation (2) ; the formula is now more reliable and you gave me an explanation about the variable 'e', but its still not explained in the text

 according to yout rebuttal letter its e=2,71.. (Euler number) or e=10 both used as base of the exponential part of the equation

5) in the line below equation (2) the Index of the first D is wrong;

should be Dw instead of DH

6) page 10 , just above Figure 8

 '.. grid points 'plan' the elevation..'

the work 'plan' seems not the right expression here - > substitute

best regards
